# Overcrowding induces fast colloidal solitons in a slowly rotating potential landscape

Eric Cereceda-López[1,2,6], Alexander P. Antonov [3,6], Artem Ryabov [4 ✉], Philipp Maass [3 ✉] & Pietro Tierno [1,2,5 ✉]

Collective particle transport across periodic energy landscapes is ubiquitously present in many condensed matter systems spanning from vortices in high-temperature superconductors, frictional atomic sliding, driven skyrmions to biological and active matter. Here we report the emergence of fast solitons propagating against a rotating optical landscape. These experimentally observed solitons are stable cluster waves that originate from a coordinated particle exchange process which occurs when the number of trapped micro-particles exceeds the number of potential wells. The size and speed of individual solitons rapidly increase with the particle diameter as predicted by theory and confirmed by numerical simulations. We show that when several solitons coexist, an effective repulsive interaction can stabilize their propagation along the periodic potential. Our experiments demonstrate a generic mechanism for cluster-mediated transport with potential applications to condensed matter systems on different length scales.

The driven motion of particles through periodic structures is intensively studied with the aim to understand and control non-equilibrium processes in biology, chemistry and physics[1–3]. Particle–particle and particle–substrate interactions yield a wide variety of phenomena as directional locking[4–6], stepwise increases of ratchet currents[7], dynamic mode locking[8], or pinning-depinning transition with a complex alternation between static and dynamic modes[9,10].

In a recent theoretical work[11] it was predicted that a collection of hard spheres driven across a periodic potential could produce solitons at high densities, i.e., stable propagating cluster waves. The predicted effect is appealing, since it allows to achieve a net transportation in dense systems even when single particles cannot surmount the energetic barriers of the potential, thus invoking for an experimental realization.

Here we experimentally show that in a rotating periodic potential with a number of particles larger than the number of potential wells, solitons emerge due to collective effects. In general, solitons are solitary waves propagating without distortion[12]. They have been found in a variety of systems from fermionic superfluids[13] and Bose-Einstein condensate[14,15], to macroscopic cracks[16], mechanical metamaterials[17–19], and ocean waves[20]. In our experiments, the solitons are composed of clusters that continuously break and reform by releasing and accepting particles via a periodic exchange particle process. Solitons are commonly known from a continuous description of the nonlinear dynamics of waves, as, e.g., in the Sine-Gordon, Korteweg-De Vries, or nonlinear Schrödinger equations and others[21]. In contrast, in our experiments the solitons emerge from many-body effects and are observed at the single particle level. Moreover, the colloidal solitons form despite of negligible inertia in the particle motion, i.e. in the limit of fully overdamped Brownian dynamics. They reach a speed much higher than that of the driving potential and even move against the driving direction.

Particle transport against an external bias is a rather unusual effect. It occurs under special circumstances, for example, in the presence of strong confinement[22–24], multi-particle interactions[25] or when the imposed potential is modulated in time[26,27]. For the soliton

[1]Departament de Física de la Matèria Condensada, Universitat de Barcelona, 08028 Barcelona, Spain. [2]Institut de Nanociència i Nanotecnologia, Universitat de Barcelona (IN2UB), 08028 Barcelona, Spain. [3]Universität Osnabrück, Fachbereich Physik, Barbarastraße 7, D-49076 Osnabrück, Germany. [4]Charles University, Faculty of Mathematics and Physics, Department of Macromolecular Physics, V Holešovičkách 2, CZ-18000 Praha 8, Czech Republic. [5]University of Barcelona Institute of Complex Systems (UBICS), 08028 Barcelona, Spain. [6]These authors contributed equally: Eric Cereceda-López, Alexander P. Antonov. ✉e-mail: artem.ryabov@mff.cuni.cz; maass@uos.de; ptierno@ub.edu

transport reported here, particle motion is still in the driving direction, but the spontaneously forming collective excitations in form of localized clusters propagate fast against the external bias.

By extending a theoretical model based on a recent prediction[11], we explain the observed speed, growth, and direction of motion of the solitons. We further strengthen our results by complementing them with Brownian dynamics simulations carried out using experimental parameters.

The underlying mechanism of soliton formation is generic and can be extended to other driven periodic systems under overcrowding conditions, such as particles flowing through porous media[28,29], vortices in high-$T_c$ superconducting heterostructures[30–32], frictional atomic sliding[33,34], driven skyrmions[35,36] ultracold atoms[37], photonic[38–40], and active matter[41–45] systems.

## Results

### Soliton observation

Our setup, schematically shown in Fig. 1a, is made of a closed fluidic cell filled with a water dispersion of spherical polystyrene particles having a diameter of $\sigma = 4\,\mu m$[46]. We confine $N$ of these particles in a rotating one-dimensional periodic potential which is created by an infrared laser beam rapidly steered across $M = 27$ equispaced positions along a circle of radius $R$. More details can be found in the Methods section. Once the laser is scanned trough the array of optical traps, it generates an effective periodic potential along the azimuthal direction with wavelength $\lambda = 2\pi R/M$, which rotates at a constant angular velocity $\omega$, Fig. 1a.

The generated optical potential strongly confines the particles in the radial direction. Indeed, particle displacements along this direction have a Gaussian distribution with a mean at the ring radius $R = 20\,\mu m$ and a standard deviation smaller than $0.01R = 0.2\,\mu m$, i.e., they are negligible compared to the particle motion along the ring. Thus, we consider particle motion to take place in a traveling-wave potential

$$U(x,t) = \frac{U_0}{2}\cos\left[\frac{2\pi}{\lambda}(x + \omega R\, t)\right],\qquad(1)$$

where $x = R\varphi$ with $\varphi$ the azimuthal angle along the ring. A single trapped colloidal particle is dragged clockwise by this translating wave. For low $\omega$, it moves at a constant tangential speed $\omega R$. By increasing $\omega$, the mean speed decreases because the particle looses its synchronized phase with the moving potential due to the viscous drag[47].

The synchronization becomes increasingly restored with larger $N$ due to an effective potential barrier enhancement by hydrodynamic interactions[48]. For $N = M$ the particle motion is fully synchronized with a mean particle velocity $\omega R$, as demonstrated in the Supplementary Movie 1.

A different type of collective excitation occurs for $N > M$. In this situation of overcrowding, after a short transitory period, the particles generate localized clusters that stably propagate along the ring without dispersion, Fig. 1b (Supplementary Movie 2). This cluster formation is not obvious in light of negligible attractive interactions between the colloidal particles in our experiments.

Let us consider the case of one extra particle, $N = M + 1$, which generates a double-occupied trap. The excess particle displaces the colloids in neighboring traps from the preferred position close to the potential minimum, forming an extended defect, which appears as an almost compact cluster composed of particles that are nearly in contact. The defect propagates as a dispersion-free solitary wave, or soliton, along the optical ring.

Surprisingly, it moves counter-clockwise along the ring, i.e., backwards against the external driving, and the speed of propagation is much faster than that of the rotating optical traps. The backward propagation occurs independent of the sense of rotation of the optical landscape, which means it is clockwise if the traps are rotated counterclockwise and vice versa. It is not caused by the viscous drag only. In the case of negligible potential barriers, the viscous drag cannot lead to backward motion, but at most to a vanishing mean displacement of the particles along the ring.

### Soliton stability and propagation

To understand the backward movement of a cluster, we must consider a wavelike cooperative movement, where the particles forming the cluster change. Specifically, this change is occurring due to particles

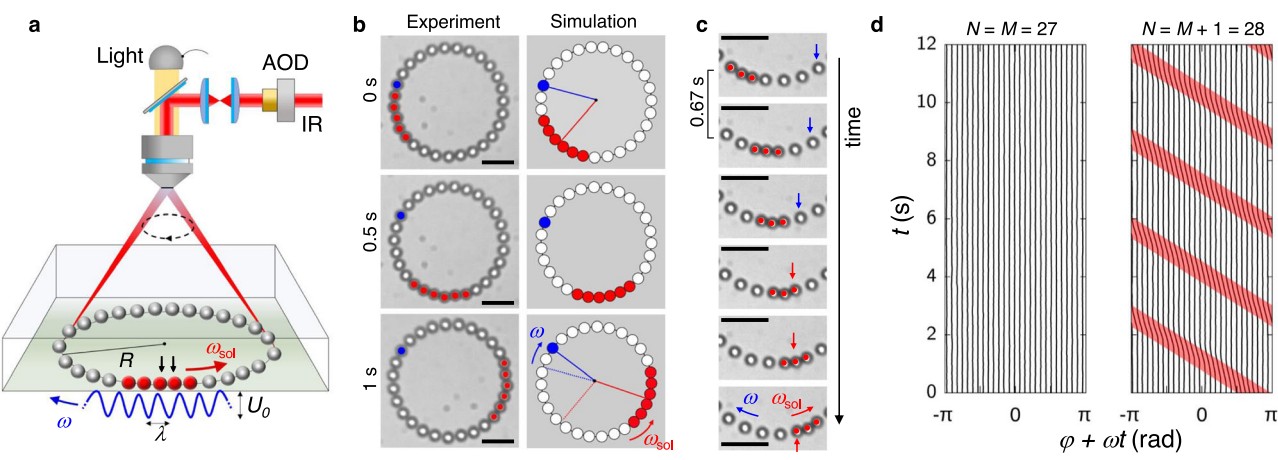

**Fig. 1 | Observation of a colloidal soliton. a** Schematic showing a ring of radius $R$ with $N$ colloidal particles of diameter $\sigma$ trapped by a rotating periodic optical potential. The potential landscape with $M$ equidistant traps of spacing $\lambda = 2\pi R/M$ and depth $U_0$ is realized using an infrared beam (IR) rapidly steered through a computer-controlled acousto-optic deflector (AOD). The traps are slowly rotated clockwise with angular velocity $\omega$. Particles highlighted in red belong to a counterclockwise propagating soliton. Black arrows indicate two particles of the soliton that occupy the same optical trap. **b** Optical microscope (first column, scale bar is 10 μm for all images) and simulation (second column) images showing a particle cluster (red) counter-propagating with angular velocity $\omega_{sol}$ against the clockwise moving optical traps. The traps drag individual particles, one dragged particle is

indicated in blue color. Parameters are $R = 20\,\mu m$, $M = 27$, $N = M + 1 = 28$, $\sigma = 0.86\lambda$, $\lambda = 4.7\,\mu m$, $\omega = 0.36\,rad\,s^{-1}$ and $U_0 = 122\,k_BT$. **c** Sequence of images separated by 0.67 s for $\sigma = 0.6\lambda$, demonstrating single particle attachments and detachments to and from a cluster in course of soliton propagation. Scale bar is 10 μm for all images. **d** Temporal evolution of the angular particle coordinates $\varphi_i(t)$ in the reference frame corotating with the traps for the completely filled system ($N = M = 27$, left) and the overcrowded system ($N = M + 1 = 28$, right) for which images are given in (**b**). In the overcrowded system, a soliton emerges as highlighted by the red region. See Supplementary Movie 1 and Movie 2. Source data are provided as a Source Data file.

detaching from the cluster at its back end and attaching at its front end, as shown in Fig. 1c. Back and front end refer to the direction of the cluster motion. However, why do particle stay together without attractive interactions and why does their cooperative movement appear as a stable soliton?

To answer these questions, let us view the motion in a frame corotating with the optical traps, for which we show particle trajectories in Fig. 1d. If the particle number $N$ is equal to the number $M$ of traps, each particle in this frame stays at the bottom of a potential well. For $N = M + 1$, coherent particle movements between the wells are present due to the soliton propagation, see the parts of the particle trajectories marked in red.

In the corotating frame, the particles are driven by a mean flow of the surrounding fluid that tries to move them clockwise, corresponding to the positive $x$-direction. The effective force of the flow-driving follows from a coordinate transformation to the comoving frame: $x' = x + \omega R t$, yielding $dx'/dt = dx/dt + \omega R$. This amounts to a constant drag force $F = \omega R/\mu$ acting on the particles in the corotating frame, which locally tilts the periodic potential. For the parameters used in our experiments, this tilting is always undercritical and the effective barrier between traps much larger than the thermal energy $k_B T$. This means that a single particle could hardly move from one optical trap to the next due to thermal excitation. In the many-particle system with filling factor one ($N = M$), where the system is perfectly ordered, the drag force $F$ does not generate sustainable motion, see Fig. 1d.

In the overcrowded system, propagating clusters of particles can occur, which are not formed by attractive interactions but are stabilized by the external forces $F^{ext}(x') = \omega R/\mu - \partial U(x')/\partial x'$, where $U(x') = (U_0/2) \cos(2\pi x'/\lambda)$ is the potential in the corotating frame. The condition on the external forces to hold $n$ particles together in an $n$-cluster is[11]

$$\frac{1}{i} \sum_{j=1}^{i} F_j^{ext} \geq \frac{1}{n-i} \sum_{j=i+1}^{n} F_j^{ext}, i = 1, \ldots, n-1. \quad (2)$$

Here, $F_j^{ext} = F^{ext}(x'_j)$ is the external force on the $j$th particle in the cluster. If inequalities (2) hold, a fragmentation of the cluster is impossible. This is because for a fragmentation to occur into a left subcluster of the first $i$ particles and a right subcluster of the $(n-i)$ remaining particles, the average force $\sum_{j=1}^{i} F_j^{ext}/i$ on the left subcluster would have to be smaller than the average force $\sum_{j=i+1}^{n} F_j^{ext}/(n-i)$ on the right subcluster.

For the particles to stay together in the cluster during its motion, the inequalities (2) must be obeyed for all points in some interval. If this interval would span a full period $\lambda$, then the cluster would move without changing its size. However, the soliton dynamics are more complex, as the size of a cluster changes within $\lambda$. This leads to different types of solitons.

## Soliton types

As the potential barrier $U_0$ is much larger than $k_B T$, formation and propagation of solitons can be understood by considering the limit of vanishing noise. Theoretically, the solitons consist of periodically repeating movements of clusters with different size in this limit[11,49], for which the equations of motions in the corotating frame are

$$\frac{dx'_i}{dt} = \mu F^{ext}(x'_i) = \omega R + \frac{\mu U_0 \pi}{\lambda} \sin\left(\frac{2\pi x'_i}{\lambda}\right). \quad (3)$$

Introducing scaled dimensionless coordinates and time, $x'_i \to y_i = x'_i/\lambda$, $t \to \lambda^2 t/(\pi \mu U_0)$, and a dimensionless driving force $f = \lambda \omega R/(\pi \mu U_0) = F\lambda/(\pi U_0)$, these equations take the form

$$\frac{dy_i}{dt} = f + \sin(2\pi y_i). \quad (4)$$

Solving Eq. (4) subject to the force conditions (2) and an initial condition with one double-occupied potential well, we find that after a transient time, periodic motions of two soliton types appear as limit cycles: an A type soliton given by two subintervals of the movements of an $n$- and $(n+1)$-cluster during one period [$n$-$(n+1)$-soliton], and a B type soliton given by four subintervals of cluster movements [$n$-$(n+1)$-$(n+2)$-$(n+1)$-soliton].

The decrease and increase of a cluster size in the sequences is by detachments and attachments of a single particle to the cluster. Exact expressions for the position of and times between attachment and detachment events are derived in Supplementary Discussion 1. Knowing these quantities, we can in particular calculate mean sizes and mean angular velocities of the solitons.

Type B solitons occur only in narrow regimes of particles sizes and driving forces. Typical solitons are of type A. Within one period of the motion of an A type soliton of core size $n$, a single particle detaches at the back end of an $(n+1)$-cluster and attaches at the front end of an $n$-cluster. Back and front end refer to the direction of cluster motion in the corotating frame, i.e., to the direction of positive $x$ for $f > 0$.

Figure 2a shows type A solitons observed in the experiments. For $\sigma = 0.67\lambda$, sequences of 3- and 2-cluster movements occur, corresponding to a 3-2-soliton. For $\sigma = 0.76\lambda$, we see the occurrence of a 4-3-soliton. The clusters in these solitons are not formed by particles that are exactly in contact, because the noise in the experiments is weak but not vanishing. Due to thermal fluctuations, particles cannot stay in contact during their motion. In our analysis of both experiments and simulations at finite noise, we identify a cluster as a sequence of neighboring particles, where the empty gap between two neighboring particles is smaller than a cutoff-distance $5 \times 10^{-3}\lambda$. The number $n$ of particles in the corresponding sequence gives the cluster size. For defining the soliton position, we select the pair of particles in the cluster whose positions are in the same optical trap, as shown in Fig. 1a.

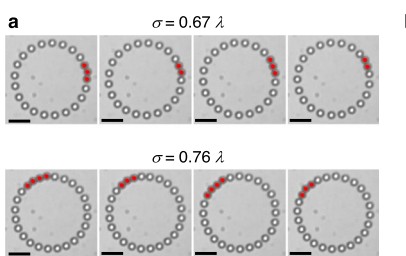
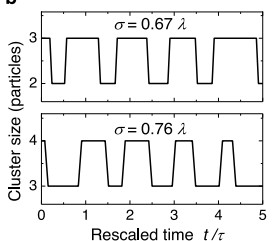
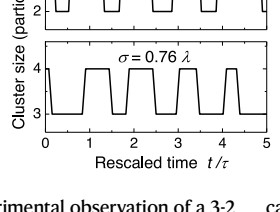
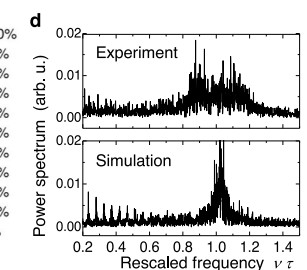
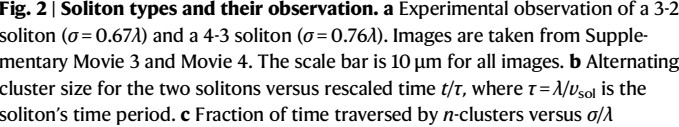

**Fig. 2 | Soliton types and their observation. a** Experimental observation of a 3-2 soliton ($\sigma = 0.67\lambda$) and a 4-3 soliton ($\sigma = 0.76\lambda$). Images are taken from Supplementary Movie 3 and Movie 4. The scale bar is 10 μm for all images. **b** Alternating cluster size for the two solitons versus rescaled time $t/\tau$, where $\tau = \lambda/v_{sol}$ is the soliton's time period. **c** Fraction of time traversed by $n$-clusters versus $\sigma/\lambda$ calculated in the limit of zero noise, represented by both the blue lines and the color maps. **d** Power spectra of soliton position versus scaled frequency $\nu\tau$ in experiments and simulation. Further parameters for all figures are given in Table 1. Source data are provided as a Source Data file.

The soliton coordinate is taken to be the center of mass position [angle coordinate $\varphi(t)$] of the two particles belonging to that pair.

Figure 2b shows the time intervals of alternating $n$- and $(n+1)$-cluster motion. The duration of two consecutive time intervals is close to the average time period $\lambda/v_{sol} = \lambda/\omega_{sol}R$, reflecting the underlying periodicity of the soliton motion. In the presence of noise, the motion is no longer perfectly periodic in time, but the power spectrum of the cluster position should still reflect the underlying periodicity by exhibiting a peak at a frequency $v_{sol}/\lambda$. Indeed, we find this peak in the spectrum in both experiments [top image in Fig. 2d] and numerical simulations [bottom image in Fig. 2d]. To filter out the trivial peak at frequency $v_{sol}/L$, we have used the relative position of the clusters to the next potential minimum in the calculation of the spectra.

In Fig. 2c we show the fraction of time a soliton stays in the $n$-cluster state in dependence of the particle diameter. These results have been calculated analytically in the zero-noise limit, see Supplementary Discussion 1. They demonstrate in particular that the clusters of the solitons become larger with increasing $\sigma$, in agreement with the experimental observations.

### Soliton size and speed

In Fig. 3a we show that the mean cluster size $\langle n \rangle$ increases rapidly with $\sigma/\lambda$, as the perturbation induced by a double-occupied trap spreads over longer distances. In the experiments (empty symbols), the wavelength $\lambda$ was varied according to $\lambda = 2\pi R/M$ with $M = 21$-27. We can estimate $\langle n \rangle$ by the following reasoning: An $n$-cluster covers a space $n\sigma$. The remaining $(N-n)$ particles not belonging to the $n$-cluster are close to potential minima and thus distributed over the length $(N-n)\lambda$. The two lengths should fill the ring of length $M\lambda = (N-1)\lambda$, yielding $(N-n)\lambda + n\sigma \simeq (N-1)\lambda$, i.e., $n\sigma \simeq (n-1)\lambda$. As this reasoning can be applied to any cluster in a soliton mode, we obtain

$$\langle n \rangle = \alpha \frac{1}{1 - \sigma/\lambda}, \tag{5}$$

where $\alpha$ is a prefactor of order unity. In Supplementary Note 4 we show that this approximate expression indeed captures the result of an exact calculation of $\langle n \rangle$ in the zero-noise limit. The experimental results for $\langle n \rangle$ in Fig. 3a can be well fitted to Eq. (5) with $\alpha = 0.83$.

Figure 3b shows the normalized mean angular soliton speed $\omega_{sol}/\omega$ as a function of the scaled particle diameter $\sigma/\lambda$, where we varied the potential wavelength $\lambda$ in the experiments, as in Fig. 3b. The observed speed raises nonlinearly with $\sigma/\lambda$, reaching a maximum value of $v_{sol} = 34.7\,\mu\text{m s}^{-1}$ for $\lambda = 4.65\,\mu\text{m}$, which is about five times higher than the speed of the optical traps moving in the opposite direction ($\omega R = 7.2\,\mu\text{m s}^{-1}$). This means that the soliton velocity relative to the optical traps is almost six times higher.

The soliton mean velocity $v_{sol}$ can be estimated by a scaling argument in the corotating frame. After the detachment and attachment of a particle, the mean distance moved by the cluster is one wavelength $\lambda$. The time needed for an attachment and detachment is of the order of the mean distance $(\lambda - \sigma)$ between two particles divided by the velocity $\omega R$. Accordingly, $v_{sol}$ should be proportional to $\lambda\omega R/(\lambda - \sigma)$, i.e. $v_{sol} = \beta\omega R/(\lambda - \sigma) = \beta\omega R\langle n \rangle$, where $\beta$ is a constant. This scaling argument is corroborated by an exact calculation of soliton velocities in the zero-noise limit outlined in Supplementary Discussion 1. In the laboratory frame, this gives

$$\omega_{sol} = \frac{v_{sol}}{R} - \omega = \frac{(\beta\alpha - 1) + \sigma/\lambda}{1 - \sigma/\lambda}\,\omega. \tag{6}$$

Figure 3b shows that the mean soliton velocity in the experiment can be well described by Eq. (6) with $\beta_{exp} = 1.02$. The simulated data can be well fitted with $\beta_{th} = 1.31$. This value is very close to the one obtained when fitting Eq. (6) to the the exact analytical calculation in the zero-

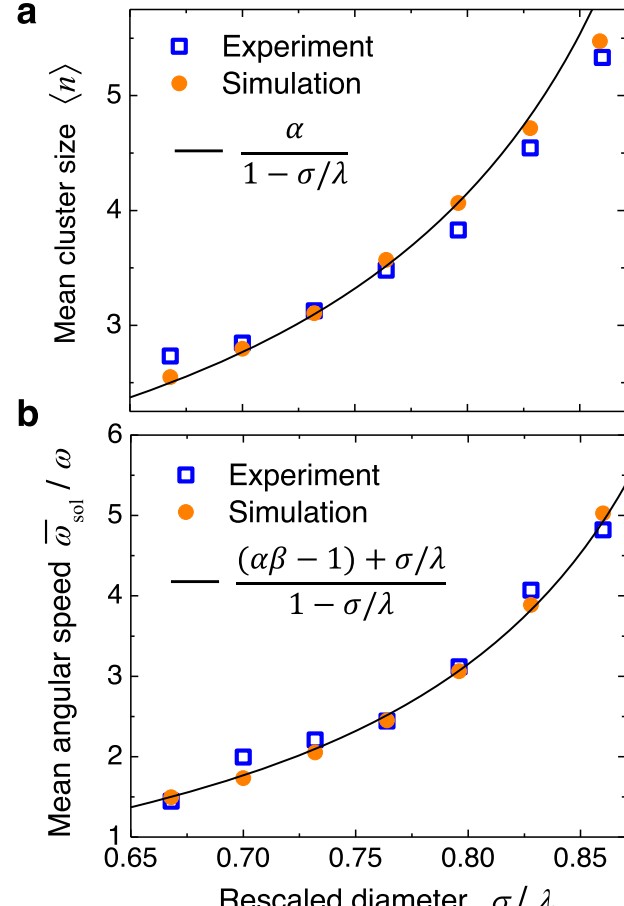

**Fig. 3 | Soliton size and speed. a** Mean size $\langle n \rangle$ of clusters forming solitons versus $\sigma/\lambda$ for $M$ potential wells and $N = M + 1$ particles ($M = 21$-27). Empty blue squares are experimental data, filled orange disks numerical simulations with corresponding parameters, and the solid line marks the theoretically predicted behavior according to Eq. (5) with $\alpha = 0.83$. **b** Normalized mean angular velocity $\bar{\omega}_{sol}/\omega$ of solitons as a function of $\sigma/\lambda$. Empty blue squares are experimental data, filled orange disks numerical simulations. The solid line marks the theoretically predicted behavior according to Eq. (6). Further parameters are given in Table 1. Source data are provided as a Source Data file.

noise limit, see Supplementary Discussion 1. The difference between $\beta_{exp}$ and $\beta_{th}$ can be explained by the fact that an ideal traveling sinusoidal wave is modeled in the simulations, while in the experiments there are always some small deviations from the ideal behavior, e.g., deviations from the exact sinusoidal form of the potential and from a purely one-dimensional particle motion. Due to these imperfections, we have rescaled the simulated soliton velocities in the corotating frame by the factor $\beta_{exp}/\beta_{th}$. The correspondingly rescaled simulated data are shown in Fig. 3a.

### Many interacting colloidal solitons

We find that the number of solitons present in our system is equal to the overcrowding, defined as the difference $N - M$. Figure 4a demonstrates the increase of the number of solitons with the overcrowding for two particle diameters. Our observations suggest that solitons tend to repel each other and propagate at a well-defined mean distance. To quantify this effect, we analyzed distributions of soliton distances.

For two solitons, the distribution $\psi(\Delta\varphi)$ of distances $\Delta\varphi(t) = |\varphi_2(t) - \varphi_1(t)|$ between the positions $\varphi_1(t)$ and $\varphi_2(t)$ of two solitons is shown in Fig. 4 for experiment (b, c) and simulation (d, e). In both cases, $\psi(\Delta\varphi)$ exhibits several peaks with a Gaussian-like envelope centered around a mean $\overline{\Delta\varphi} \simeq \pi$. Such fine structure appears due to the fact that the solitons have preferential positions close to the

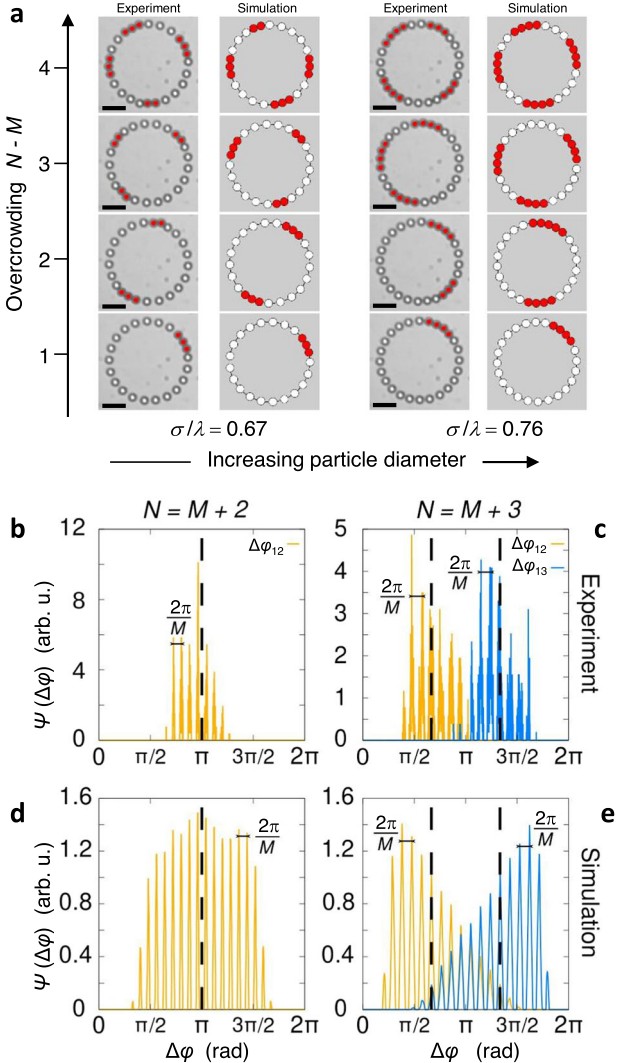

**Fig. 4 | Soliton-soliton interaction. a** Number of solitons at different over-crowdings $N-M$ and two ratios $\sigma/\lambda$. First (second) column are experimental (simulation) images with solitons highlighted in red color. Supplementary Movie 5 shows one particular case ($\sigma/\lambda = 0.67$, $N-M = 3$). The scale bar is 10 μm for all images. **b**–**e** Distributions $\psi$ of the angle distance $\Delta\varphi$ between two (**b**, **d**) and three (**c**, **e**) solitons from experiments (**b**, **c**) and numerical simulations (**d**, **e**). For two solitons, $\Delta\varphi = \Delta\varphi_{12}(t) = |\varphi_2(t) - \varphi_1(t)|$ (yellow lines). For three solitons, $\Delta\varphi_{12}(t) = \varphi_2(t) - \varphi_1(t)$ (yellow lines) and $\Delta\varphi_{13}(t) = \varphi_3(t) - \varphi_1(t)$ (blue lines), where the positions of the solitons are ordered according to $\varphi_1(t) < \varphi_2(t) < \varphi_3(t)$. In the left graphs, the dashed vertical lines mark the distance $\Delta\varphi = \pi$ and in the right graphs $\Delta\varphi = 2\pi/3$ and $\Delta\varphi = 4\pi/3$. Parameters of the experiments (simulations) are given in Table 1. Source data are provided as a Source Data file.

**Table 1 | Parameters in the experiments and simulations**

| Figures | $M$ | $N-M$ | $\sigma/\lambda$ | $U_O/k_BT$ |
|---|---|---|---|---|
| 1b, 1d | 27 | 1 | 0.86 | 122 |
| 1c | 19 | 1 | 0.60 | 211 |
| 2a, 2b, 2d | 21 | 1 | 0.67 | 179 |
| 2a, 2b | 24 | 1 | 0.76 | 158 |
| 2c | 22 | 1 | 0.6–0.85 | 176–124 |
| 3a, 3b | 21 | 1 | 0.67 | 179 |
| | 22 | 1 | 0.70 | 178 |
| | 23 | 1 | 0.73 | 165 |
| | 24 | 1 | 0.76 | 158 |
| | 25 | 1 | 0.80 | 140 |
| | 26 | 1 | 0.83 | 128 |
| | 27 | 1 | 0.86 | 122 |
| 4a | 21 | 1–4 | 0.67 | 179 |
| | 24 | 1–4 | 0.76 | 142 |
| 4b | 25 | 2 | 0.8 | 140 |
| | 21 | 3 | 0.67 | 179 |

$M$ is the number of optical traps, $N-M$ is the overcrowding, and $U_O/k_BT$ is the potential barrier between optical traps in units of the thermal energy. The wavelength is $\lambda = 2\pi R/M$. Parameters that are the same in all experiments are $R = 20$ μm, $\sigma = 4$ μm, $\omega = 0.36$ rad s$^{-1}$, and $D = 0.1295$ μm$^2$ s$^{-1}$.

## Discussion

We have experimentally observed solitary cluster waves in a highly crowded system of driven Brownian particles along a periodic potential. The solitons are robust, their number can be controlled by the overcrowding, and they propagate against the rotation direction at a speed up to six times higher than the angular velocity of the rotating wells. Viewed in a frame comoving with the potential wells, the experiment realizes particle motion through a periodic potential driven by a constant drag force. The solitons then move in the force direction and provide a formidable way to transport matter at the microscale in confined space.

Cluster-mediated transport realized in our work may appear similar to the dynamics of kinks and anti-kinks predicted by the Frenkel-Kontorova (FK) model[50], as, for example, seen in nanotribology[33,34] and diffusion of crowdions[51,52] and voidions[53,54]. In recent experimental realizations on the colloidal length scale, kinks were produced by sliding electrostatically repulsive microspheres across a static, two-dimensional potential[55,56]. The mechanism behind the kink formation in the FK model appears, however, to be fundamentally different from the mechanism of soliton formation reported here, which is given by the conditions in Eq. (2) between the external forces. Due to these conditions, particles can keep together during cluster propagation. This mechanism leads to cluster waves that, once excited, are robust and persist during all experimental time.

Cluster-mediated transport is widespread in many artificial and biological systems including, for example, microfluidic channels or vein networks. The rich physics unveiled in an overcrowded single-file system driven across a periodic potential opens different perspectives for further research.

## Methods

### Experimental system

Our colloidal suspension consists of spherical polystyrene particles having a diameter of $\sigma = 4$ μm (CML, Molecular Probes, 4% w/v in water) that are dispersed in deionized water (MilliQ). This suspension is confined within a fluidic cell, Fig. 1a, which is composed of two coverslips separated by ≈ 100 μm and placed on the stage of a custom-

potential minima, while other positions are less likely. Accordingly, the distance distribution shows peaks separated by $2\pi/M$. The localized envelope points to an effective repulsive soliton-soliton interaction, which tends to keep the soliton positions at a maximal distance $\pi$. Note that the difference between the experimental (Fig. 4b, c) and simulation data (Fig. 4d, e) is only in the widths of the angle distributions, but not in the mean value, which demonstrates the presence of an effective soliton–soliton interaction.

For steady states characterized by three propagating solitons, their position can be ordered at each time $t$ such that $\varphi_1(t) < \varphi_2(t) < \varphi_3(t)$. Thus we show in Fig. 4c, e the distributions $\psi(\Delta\varphi_{12})$ and $\psi(\Delta\varphi_{13})$ of the distances $\Delta\varphi_{12} = \varphi_2(t) - \varphi_1(t)$ and $\Delta\varphi_{13} = \varphi_3(t) - \varphi_1(t)$. Here the envelopes of $\psi(\Delta\varphi_{12})$ and $\psi(\Delta\varphi_{13})$ are centered close to $2\pi/3$ and $4\pi/3$, respectively.

built optical microscope. The particles are left sediment close to the bottom of the cell due to density mismatch, and float there showing a small diffusion coefficient $D = 0.1295\ \mu m^2 s^{-1}$, as measured from the mean squared displacement in absence of the optical potential. The system dynamics are visualized by using a Nikon 40 × microscope objective (plan Apo) illuminated by a light emitting diode, while video recording is performed at 60 Hz with a complementary metal oxide camera (Ximea MQ003MG-CM).

We create a rotating one-dimensional periodic potential by passing an infrared laser beam (Manlight ML5-CW-P/TKS-OTS) through a pair of acousto-optic deflectors (AA Optoelectronics DTSXY-400-1064). The beam is characterized by a wavelength 1064 μm and power $P = 3$ W. The AODs have an input frequency in the range from 60 to 90 MHz. It is produced via a two-channel radio frequency wave generator (DDSPA2X-D431b-34) which is addressed by a digital output card (National Instruments cDAQ NI-9403) with a refresh frequency of 150 kHz. More technical details on the experimental system can be found in a related work[47], which investigates the dynamics of a number of particles in the low filling regime, $N < M$,[46].

### Numerical simulations
In the experiments, the particles perform a Brownian motion in the time-dependent optical potential generated by the rotating laser beam. As shown in ref. [47], these dynamics can be described by that of hard-spheres with diameter $\sigma$, where the center of mass position $\boldsymbol{r}_i$ of each particle $i$ moves according to the Langevin equation

$$\frac{d\boldsymbol{r}_i}{dt} = -\mu\boldsymbol{\nabla}U_{opt}(\boldsymbol{r}_i,t) + \boldsymbol{\zeta}_i(t). \tag{7}$$

Here $U_{opt}(\boldsymbol{r},t)$ is the time-dependent optical potential, $\mu = D/k_B T$ is the particle mobility, and $\boldsymbol{\zeta}_i(t)$ are Gaussian white noise processes with $\langle\boldsymbol{\zeta}_i(t)\rangle = 0$ and $\langle\zeta_{i\alpha}(t)\zeta_{j\beta}(t')\rangle = 2D\delta_{ij}\delta_{\alpha\beta}\delta(t-t')$. Because the particle confinement in the radial direction is very strong and the radius of curvature is much greater than the mean distance between the colloidal particles, the motion can be restricted to one dimension in a traveling-wave potential $U(x,t) = (U_0/2)\cos(2\pi x/\lambda + M\omega t)$ with $x = R\varphi$, where $\varphi$ is the azimuthal angle[47,57,58]. Note that $\omega$ is the angular velocity of the rotation of the optical traps, which implies that the traveling wave has the frequency $M\omega$ for $M$ traps. The impact of hydrodynamic interactions is effectively tackled by the recently discovered potential barrier enhancement[48] and the condition $|\boldsymbol{r}_i - \boldsymbol{r}_j| \geq \sigma$ implied by the hard-sphere interaction is treated by the method proposed in ref. [59] and additionally by a recently developed cluster algorithm[60]. The two simulations methods gave the same results.

## Data availability
The data that support the findings of this study are available from the corresponding author upon request. Parts of the originally submitted manuscript have been incorporated in a doctoral thesis[46] available at http://hdl.handle.net/10803/688857. Source data are provided with this paper.

## Code availability
All computer codes are available from the corresponding authors upon request.

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

## Acknowledgements

This project has received funding from the European Research Council (ERC) under the European Union's Horizon 2020 research and innovation programme (grant agreement no. 811234). A.A., A.R., and P.M. gratefully acknowledge financial support by the Czech Science Foundation (Project No. 20-24748J) and the Deutsche Forschungsgemeinschaft (Project No. 432123484). P.T. acknowledges support from the Ministerio de Ciencia e Innovació (Project No. PID2022-137713NB-C22), the Agència de Gestió d'Ajuts Universitaris i de Recerca (Project No. 2021 SGR 00450) and the Generalitat de Catalunya under Program "ICREA Acadèmia".

## Author contributions

E.C. performed the experiments and analyzed the experimental data. A.A. performed numerical simulations, analyzed experimental data, and derived analytical formulas presented in the supplementary information. A.R., P.M., and P.T. supervised the work. All authors discussed the results and commented on the manuscript at all stages.

## Competing interests

The authors declare no competing interests.
