## [Peer Review File · Nature Communications]

nature portfolio

Peer Review FileResponse to reviewer comments, first round:

Reviewer #1 (Remarks to the Author):

This submission to Nat. Comm. reports an experimental/theoretical study on the motion of Brownian particles in a rotating periodic energy landscape which is created by a rotating focused laser beam. At incommensurate conditions, i.e., when the particle number is larger than the number of traps, the authors observe dispersion-free solitons running against the direction of the rotating laser field. Furthermore, it is shown that the velocity of solitonic excitations can be much faster than the velocity of the rotating wells and that their velocity depends on the particle diameter. At sufficiently high overcrowding, several solitons are formed which do not annihilate but interact via a short-ranged repulsive interaction. The experimental observations are in excellent agreement with a theoretical model.

This is a well-written and sound paper which reports on cute and very clean experiments on an interesting subject. The simulations corroborate the findings in an excellent manner, and I agree with the relevance of the results to many other areas of science.

However, to my opinion the main issue with the paper is, that the discussed physics is not that new as claimed by the authors. In fact, the situation considered by the authors is just a one-dimensional Frenkel-Kontorova model which has been developed back more than 80 years ago. Such models were discussed in the realm of nanotribology since several decades in simulations but there exist also several experiments with atoms, ions and even colloidal systems. Some of those experiments are on one-dimensional FK systems, others are two-dimensional. In those studies, essentially identical results observed by the authors are reported even though the solitons are often called kinks (or antikinks).

Within the FK framework the experimental observations of the authors are immediately understood, so the results are not surprising.

The force that makes it "surprisingly" move backward is simply the viscous drag. With the wells rotating clockwise, friction drag from the fluid pulls all particles against the advancing wells, thus applying a counterclockwise force/torque to all particles. However, of all 28 particles, 27 are nominally locked to a well, so they cannot slip. The extra particle, surrounded by its distortion "cloud", forms the soliton, which is freer to move. Not entirely free actually, it is also pinned by a weak Peierls-Nabarro barrier, but that is far smaller than the single-particle barrier, so the viscous drag is sufficient to dislodge it from the wells. This is all not really new (both from a theoretical and experimental side) even though the authors have not given credit to such framework.

That the solitonic velocity varies with the amount particle size and amount of overfilling is easily explained within the FK model as well. Depending on the incommensurability, the lateral extension of solitons increases which reduces their Peierls-Nabarro barrier. The observations of the authors are in full agreement with this.

What is really new (at least from an experimental side) is the interaction of solitons which I have not seen in experiments before. I am not sure, however, whether this is sufficient to warrant publication in Nat. Comm. In any case, the authors should make clear in what regard their observations are different compared what has been reported in the literature before.

Reviewers #2 and #3, who co-reviewed the work (Remarks to the Author):

Using experiments, mathematical analysis, and simulations the authors investigate the emergence of cluster solitons in an overcrowded system of colloidal particles coupled to a rotating periodic potential. They find that the clusters can propagate against the driving direction, that the speed of clusters increases with increasing particle diameter, and that the number of clusters that forms is

proportional to the degree of overcrowding.

The results are sound and the presentation is reasonably clear; however, the experimental results and corresponding scaling arguments regarding the average cluster size and speed, while interesting, do not represent a significant advance in the field. The most substantial portions of the theory, along with supporting simulations, were already published by a subset of the authors in Ref. 45. The experiments and simulations in the present work, while of value, do not add significantly new insight. Thus this work does not meet the criteria of novelty and importance for the broad readership of Nature Communications, and is better suited for a more specialized journal.

I have some specific comments:

1. The presentation of the paper needs improvement. A clear connection to the work published in Ref. 45 should appear in the introduction. The significance of the work should also be articulated better. The field of solitons is very old, and merely observing that solitons are a generic mechanism for transport applicable to many areas is far from being an interesting new result. The authors should explain what makes solitons in this particular system of special interest.
2. Did the authors check whether the scan direction of their rotating optical potential makes any difference? In other words, if the scan is rotated the other way, do the same results appear but in the opposite direction?
3. Fig. 2(a) is essentially the same result that appeared as Fig. 3 in Ref. 45, so at minimum this should be acknowledged. Fig. 2(b) showing the "type B" solitons does not seem to be well connected to the rest of the results. The text notes that type B solitons are only predicted from theory to occur for a small range of parameters. What is this range of parameters? The implication seems to be that type B solitons are not observed in either experiment or simulation. Is this correct? If so, the type B solitons seem somewhat irrelevant. The authors should either better explain how type B solitons could be observed or perhaps it would be better to omit the type B solitons, since the description of the type B solitons as it currently stands does not make a useful contribution to the results in the paper.
4. How do the results shown in Fig. 2(e) compare with the experiments and simulations? Space spent discussing type B solitons could perhaps be better employed in giving a more detailed description of the results in Fig. 2(e).
5. Just after Eq. (4) the authors write "Solving Eq. (4) subject to the force conditions..." What do the authors mean by "force conditions?"
6. In Eq. (5), is there any physical interpretation of the prefactor α , or is it possible to provide any description of the significance of the value of α ? Is the equation with $\alpha = 1$ exact in the zero noise limit?
7. Why is simulation data omitted in Fig 3(b)? Also, what value of α is used for the solid line in Fig. 3(b)?
8. The interacting soliton regime could have been an interesting new angle, but the results shown in Fig. 4 seem very preliminary and are not well quantified. Is there a reason for the apparent discrepancy between experiment and simulation in Fig. 4(b)? In particular, the distributions are much broader in the simulation and the peaks do not match in the $N-M = 3$ case.

Reviewer #4 (Remarks to the Author):

The authors describe a movement of a cluster of colloids in an overpopulated trap, in the counter-rotated direction in theory and experiment. I think this paper is interesting and suitable for Nature Communication, but would ask the authors to clarify several parts of their manuscript before I

recommend it for publication:

- 1) Can the authors specify somewhat more specific what they define as a soliton/solitary wave? For example, the authors write: "This cluster formation is not obvious in light of negligible attractive interactions between the colloidal particles in our experiments." – What balances the dispersion in their case? If there is no attractive interaction can it really be called a soliton? Do the authors have references where this terminology has been used already?
- 2) Can the authors specify, how they define the size of the cluster? – Specifically, what defines which particles are colored in red in the figures?
- 3) Figure 1a uses dark red, which is later not used anymore. This might be confusing for the reader.
- 4) I do believe that the paper would benefit tremendously from an improved presentation of Fig. 2 in a more compressed and intuitive way? – There are several redundancies in this figure. Maybe the authors can find some way to compress the figure with a simpler design. Especially Fig. 2a,b were hard to understand (for me).
- 5) Can the experiment be understood completely classically?

Minor comments:

- I would give the reason why particle displacement in the radial direction can be neglected in the main part of the text. Additional information can then be given in the methods.
- I would refrain from using the wording "phase difference" to describe the distance, as the first terminology is often used in the quantum mechanical context with a completely different meaning.

Response Letter to Reviewers

We thank all the Reviewers for their thorough reading of our manuscript and their suggestions/criticisms. Our point-by-point responses and the corresponding changes to the manuscript are described below.

Response to Reviewer #1

Reviewer: *This submission to Nat. Comm. reports an experimental/theoretical study on the motion of Brownian particles in a rotating periodic energy landscape which is created by a rotating focused laser beam. At incommensurate conditions, i.e., when the particle number is larger than the number of traps, the authors observe dispersion-free solitons running against the direction of the rotating laser field. Furthermore, it is shown that the velocity of solitonic excitations can be much faster than the velocity of the rotating wells and that their velocity depends on the particle diameter. At sufficiently high overcrowding, several solitons are formed which do not annihilate but interact via a short-ranged repulsive interaction. The experimental observations are in excellent agreement with a theoretical model.*

This is a well-written and sound paper which reports on cute and very clean experiments on an interesting subject. The simulations corroborate the findings in an excellent manner, and I agree with the relevance of the results to many other areas of science.

Authors: We thank the Reviewer for this overall positive assessment and for recognizing the relevance of these results to other areas of science.

Reviewer: *However, to my opinion the main issue with the paper is, that the discussed physics is not that new as claimed by the authors. In fact, the situation considered by the authors is just a one-dimensional Frenkel-Kontorova model which has been developed back more than 80 years ago. Such models were discussed in the realm of nanotribology since several decades in simulations but there exist also several experiments with atoms, ions and even colloidal systems.*

Authors: There are fundamental differences between our system and the FK model.

The FK model, whether under- or overdamped, considers a series of point particles coupled by springs (harmonic interactions in the basic model) and moving, due to an external force, through a fixed, periodic potential. By contrast, our system is made of colloidal particles that interact via hard-sphere interactions. The emergence of stable solitons occurs under overcrowding, i.e. when the number of traps is larger than that of potential wells. The solitons manifest themselves in the formation of particle clusters, where the particle size plays a crucial role.

The mechanism behind the cluster formation is fundamentally different from the kink formation in the FK model, namely it is given by conditions between the external forces acting on the particles forming the clusters, see Eq. (2) in the main text. To the best of our knowledge, these conditions cannot be deduced from or linked to the FK model.

Reviewer: *Some of those experiments are on one-dimensional FK systems, others are two-dimensional. In those studies, essentially identical results observed by the authors are reported even though the solitons are often called kinks (or antikinks).*

Authors: These results are not identical, because of the fundamentally different origin of soliton formation.

In the experimental realization on the colloidal length scale, see, for example, the supporting videos in *Thomas Bohlein et al., Nature Materials 11, 126 (2012)*, the kinks appear and disappear since they change size/shape during transport. This is also due to the two-dimensional nature of the system that induces kink dispersion. By contrast, our solitons persist during the experimental observation time, i.e. they do not decay.

If we further consider the mechanism of propagation of kinks [see for example, Fig. 3(c,d) in *Nature Materials 11, 126 (2012)*], the kinks move along the force directions as extra particles, the antikink are “holes” propagating against the force directions. By contrast, our solitons are clusters that exchange particles and propagate against the rotation of the potential with a speed much higher than the external driving (speed of optical trap rotation).

Reviewer: *The force that makes it "surprisingly" move backward is simply the viscous drag. With the wells rotating clockwise, friction drag from the fluid pulls all particles against the advancing wells, thus applying a counterclockwise force/torque to all particles. However, of all 28 particles, 27 are nominally locked to a well, so they cannot slip. The extra particle, surrounded by its distortion "cloud", forms the soliton, which is freer to move. Not entirely free actually, it is also pinned by a weak Peierls-Nabarro barrier, but that is far smaller than the single-particle barrier, so the viscous drag is sufficient to dislodge it from the wells. This is all not really new (both from a theoretical and experimental side) even though the authors have not given credit to such framework.*

Authors: A viscous drag and the size of the potential barrier are not sufficient to explain what we understand as “moving against the rotating landscape”.

Perhaps we have not been clear enough about this point. What we mean is that the solitons move in a direction opposite to the trap rotation in *the laboratory frame*. Consider, for example, a clockwise rotation of the optical traps. The viscous drag on a particle or particle cluster can at most (in the case of zero potential barrier) lead to a situation where its mean position remains constant in the laboratory frame. However, the solitons propagate with high speed counter-clockwise in the laboratory frame.

We have pointed this out now in the manuscript, see the end of the section “Soliton observation”:

“The backward propagation occurs independent of the sense of rotation of the optical landscape, which means it is clockwise if the traps are rotated counter-clockwise. It is not caused by the viscous drag only. In the extreme case of negligible potential barriers, the viscous drag cannot lead to backward motion, but at most to a zero-mean displacement of the particles along the ring.”

The motion against the rotating landscape arises due to the particle exchange processes during the propagation of the A and B type solitons.

Reviewer: *That the solitonic velocity varies with the amount particle size and amount of overfilling is easily explained within the FK model as well. Depending on the incommensurability, the lateral extension of solitons increases which reduces their Peierls-Nabarro barrier. The observations of the authors are in full agreement with this.*

Authors: Because the mechanism of soliton formation is fundamentally different from the FK model, the variation of soliton velocities with particle size cannot be explained by this model.

There is no simple monotonic dependence of potential barriers with soliton size. The major cause of the increase of soliton velocities is the increase of the soliton size with increasing hard sphere diameter, which is not related to potential barriers.

Reviewer: *What is really new (at least from an experimental side) is the interaction of solitons which I have not seen in experiments before. I am not sure, however, whether this is sufficient to warrant publication in Nat. Comm. In any case, the authors should make clear in what regard their observations are different compared what has been reported in the literature before.*

Authors: We agree with the Reviewer that we should make clearer why our observations are different from the kinks observed/predicted within the context of the FK model. Thus, we have added the following text in the discussion session (page 6, column 1):

“Cluster-mediated transport realized in our work may appear similar to the kinks and anti-kinks predicted by the Frenkel Kontorova (FK) model within the context of nanotribology [49, 50; New Reference 1]. In recent experimental realizations on the colloidal length scale, kinks were produced by sliding electrostatically repulsive microspheres across a static, two-dimensional potential [New Reference 2, New Reference 3]. The mechanism behind the kink formation in the FK model is, however, fundamentally different from the mechanism of soliton formation reported here, which is given by the conditions (2) between the external forces. Due to these conditions, particles can keep together during cluster propagation. This different mechanism leads to cluster waves that, once excited, are extremely robust and persist during all experimental time.

[New Reference 1] O. M. Braun and Y. S. Kivshar, Phys. Rep. 306, 1–108 (1998).

[New Reference 2] T. Bohlein, J. Mikhael, and C. Bechinger, Nat. Mater. 11, 126 (2012).

[New Reference 3] T. Brazda, C. Julya, and C. Bechinger, Soft Matter 13, 4024 (2017).

We provide a marked copy of the revised manuscript, where the changes described above are marked in blue.

Response to Reviewers #2 and #3

Reviewers: *Using experiments, mathematical analysis, and simulations the authors investigate the emergence of cluster solitons in an overcrowded system of colloidal particles coupled to a rotating periodic potential. They find that the clusters can propagate against the driving direction, that the speed of clusters increases with increasing particle diameter, and that the number of clusters that forms is proportional to the degree of overcrowding.*

The results are sound and the presentation is reasonably clear; however, the experimental results and corresponding scaling arguments regarding the average cluster size and speed, while interesting, do not represent a significant advance in the field. The most substantial portions of the theory, along with supporting simulations, were already published by a subset of the authors in Ref. 45. The experiments and simulations in the present work, while of value, do not add significantly new insight. Thus this work does not meet the criteria of novelty and importance for the broad readership of Nature Communications, and is better suited for a more specialized journal.

Authors: The main concern of the Reviewers is related to the presence of a previous work (Ref. 45 old version, now Ref. 11). This was a theoretical manuscript which predicted the existence of solitons in a system of hard spheres in a periodic potential. The present study is an experimental demonstration that these solitons occur.

We respectfully disagree that the presence of a theoretical work reduces the impact of an experimental observation. Following such reasoning, one could argue that the experimental observation of the Bose-Einstein condensation by Cornell, Wieman and Ketterle (Nobel Prize in 2001) would have been unnecessary since Bose already predicted the phenomenon theoretically in 1924 (or that the observations on Brownian motion by Perrin, or the observation of gravitational waves by Weiss, Barish and Thorne were unnecessary, etc.).

Moreover, while the theory presented in our work may be considered as an extension of that in Ref. 11, the corresponding experiments, simulations, and data analysis of single and many solitons are completely new and go beyond what was presented in Ref. 11. We even show new emerging features of the experimental system that were not contemplated in Ref. 45. These are the speed of the cluster waves that can be effectively measured and reaches up to 34.7 micron/s, which is about five times higher than the speed of the optical traps moving in the opposite direction, and the fact that different solitons can coexist, displaying an effective, repulsive interaction. The direct measurements of physical quantities together with the direct observation of the cluster exchange mechanisms at the single-particle level highlight the impact and importance of our work.

Reviewer: *I have some specific comments:*

1. The presentation of the paper needs improvement. A clear connection to the work published in Ref. 45 should appear in the introduction. The significance of the work should also be articulated better. The field of solitons is very old, and merely observing that

solitons are a generic mechanism for transport applicable to many areas is far from being an interesting new result. The authors should explain what makes solitons in this particular system of special interest.

Authors: We agree with the Reviewers that we should give more credit to Ref. 45 (now Ref. 11 in the new manuscript) since it represents the theoretical prediction of some of the observations reported in this work. We have placed in the introduction the following sentence that makes a clear connection to the previous work:

“In a recent theoretical work [11] it was predicted that a collection of hard spheres driven across a periodic potential could produce solitons at high densities, i.e. stable propagating cluster waves. The predicted effect is appealing, since it allows to achieve a net transportation in dense systems even when single particles cannot surmount the energetic barriers of the potential, thus invoking for an experimental realization. Here we experimentally show...”

The existence of solitons is known for a long time in the nonlinear dynamics of waves, as e.g. for the Sine-Gordon, Korteweg-De Vries, nonlinear Schrödinger equations etc. Our study is important because of two novel aspects: First, the solitons are observed experimentally at the microscale of individual particles, including identification of the underlying microscopic particle exchange processes. Second, the mechanism of our solitons is new, resulting from a collective many-body effect.

We emphasize this point in the text and write on page 1, column 2 the following:

“Solitons are commonly known from a continuous description of the nonlinear dynamics of waves, as, e.g., in the Sine-Gordon, Korteweg-De Vries, or nonlinear Schrödinger equations and others [New Reference]. In contrast, in our experiments the solitons emerge from many-body effects and are observed at the single particle level. Moreover, the colloidal solitons form despite of negligible inertia in the particle motion, i.e. in the limit of fully overdamped Brownian dynamics.

[New Reference] M. J. Ablowitz, Nonlinear Dispersive Waves: Asymptotic Analysis and Solitons, Cambridge Texts in Applied Mathematics (Cambridge University Press, Cambridge, 2011).

2. *Did the authors check whether the scan direction of their rotating optical potential makes any difference? In other words, if the scan is rotated the other way, do the same results appear but in the opposite direction?*

Authors: We have checked that the phenomenon is invariant upon change of the rotation direction. The same type of solitons emerges clockwise for counter-clockwise rotating optical traps and vice versa. We have commented on this point, and write at the end of the section “Soliton observation” (page 2, second column):

“The backward propagation occurs independent of the sense of rotation of the optical landscape, which means it is clockwise if the traps are rotated counter-clockwise.”

3. Fig. 2(a) is essentially the same result that appeared as Fig. 3 in Ref. 45, so at minimum this should be acknowledged. Fig. 2(b) showing the "type B" solitons does not seem to be well connected to the rest of the results. The text notes that type B solitons are only predicted from theory to occur for a small range of parameters. What is this range of parameters? The implication seems to be that type B solitons are not observed in either experiment or simulation. Is this correct? If so, the type B solitons seem somewhat irrelevant. The authors should either better explain how type B solitons could be observed or perhaps it would be better to omit the type B solitons, since the description of the type B solitons as it currently stands does not make a useful contribution to the results in the paper.

Authors: Ideal periodic sequences of motions of clusters with different size occur in the zero-noise limit. The A and B type solitons are representing such periodic sequences. At finite temperature, the thermal noise perturbs the ideal periodicity.

The type B solitons are seen in the simulations and they occur in a range of small particle diameters σ , which can be calculated based on the parameters given in Table 1 and the analytical results given in the SI. For example, for the rotation frequency ω in the experiment, the 2-1-2-3 soliton occurs for $0.56 \lambda < \sigma < 0.578 \lambda$, the 3-2-3-4 soliton for $0.54 \lambda < \sigma < 0.56 \lambda$ and the 4-3-4-5 soliton for $0.761 \lambda < \sigma < 0.764 \lambda$. We see indications of the occurrence B type solitons also in the experiment, see the videoS2, where a sequences of clusters with sizes 5, 4, 5 and 6 occurs corresponding to a 5-4-5-6 soliton.

Hence, the type B solitons are observed and they are relevant. Nevertheless, as we explain the different soliton modes in detail in the SI, we have removed Figs. 2(a) and 2(b) from the manuscript and shortened the main text related to these figures.

4. How do the results shown in Fig. 2(e) compare with the experiments and simulations? Space spent discussing type B solitons could perhaps be better employed in giving a more detailed description of the results in Fig. 2(e).

Authors: Figure 2(e) shows an exact theoretical result in the zero-noise limit, as described in the figure caption and in the last paragraph of the section "Soliton types". It is relevant in connection with the experiment to demonstrate that the clusters of the solitons become larger with increasing particle diameter, in agreement with the experimental observations.

5. Just after Eq. (4) the authors write "Solving Eq. (4) subject to the force conditions..." What do the authors mean by "force conditions?"

Authors: We have added the reference to the force conditions, which are given in Eq. (2).

6. In Eq. (5), is there any physical interpretation of the prefactor α , or is it possible to provide any description of the significance of the value of α ? Is the equation with $\alpha = 1$ exact in the zero noise limit?

Authors: The exact result in the zero-noise limit is quite involved, as it requires to calculate the cycle duration times of the solitons, see Eqs. (S15) and (S16) in the SI. In

Fig. S2b of the SI we show that the reasoning yielding the expression Eq. (5) in the main text yields a very good approximate description of the soliton velocities. When fitting Eq. (5) to the exact result Eq. (S20) in the zero-noise limit, we obtain $\alpha = 1.11$. For the results in Fig. 3b at finite temperature, a slightly lower value $\alpha = 0.83$ fits the data.

7. Why is simulation data omitted in Fig 3(b)? Also, what value of alpha is used for the solid line in Fig. 3(b)?

Authors: The simulation data in Fig. 3(b) have been omitted for simplicity. As described in the SI, the soliton velocities can be calculated analytically in the zero-noise limit and the simulation results for the low temperature used in the experiment are in excellent agreement with the theoretical prediction. In the experiments, however, there are always small deviations from the ideal behaviour (deviations from the exact sinusoidal form of the potential, deviations from a purely one-dimensional particle motion). As for our reasoning of Eq. (6) in the main text, giving a very good approximation to the exact analytical derivation in the SI, this yields a soliton velocity being proportional to $\omega R \langle n \rangle$. Taking into account a proportionality factor β , we can take into account the imperfections in the measurements by rescaling the theoretical value $\beta = 1.31$ for the ideal situation to the $\beta = 1.02$ in the experiment. We have included the corresponding simulation data in Fig. 3b and have explained this point, see the revised text before and after Eq. (6).

8. The interacting soliton regime could have been an interesting new angle, but the results shown in Fig. 4 seem very preliminary and are not well quantified. Is there a reason for the apparent discrepancy between experiment and simulation in Fig. 4(b)? In particular, the distributions are much broader in the simulation and the peaks do not match in the $N-M = 3$ case.

Authors: The seeming discrepancy is only in the widths of the angle distributions, not in the mean value, which are most relevant for demonstrating the soliton-soliton interaction. As for the widths, the simulated data have been obtained from 10^5 independent stochastic particle trajectories to obtain good statistics. When determining the angle distribution from a much smaller number of trajectories and an observation time as in the experiments, we obtain much smaller widths, which are comparable to that obtained from the measurement. The following figure shows an example:

We now explain this point in the text as follows:

“Note that the difference between the experimental and simulation data in Fig.4b is only in the widths of the angle distributions, but not in the mean value, which demonstrates the presence of an effective soliton-soliton interaction.”

We provide a marked copy of the revised manuscript, where the changes described above are marked in blue.

Response to Reviewer #4

Reviewer: *The authors describe a movement of a cluster of colloids in an overpopulated trap, in the counter-rotated direction in theory and experiment. I think this paper is interesting and suitable for Nature Communications,*

Authors: We are pleased that the Referee finds our paper interesting and considers it to be suitable for *Nature Communications*.

Reviewer: *1) Can the authors specify somewhat more specific what they define as a soliton/solitary wave? For example, the authors write: “This cluster formation is not obvious in light of negligible attractive interactions between the colloidal particles in our experiments.” – What balances the dispersion in their case? If there is no attractive interaction can it really be called a soliton? Do the authors have references where this terminology has been used already?*

Authors: We use “solitons” to denote dispersionless waves of particle clusters. In a coarse-grained description, these cluster waves represent density waves that propagate without dispersion, like in the common continuum description of solitary waves, as, e.g., found for the Sine-Gordon, Korteweg-De Vries, nonlinear Schrödinger equations etc. Our solitons are novel: They are observed experimentally on the microscale of individual particles, which includes identification of the underlying microscopic particle exchange processes. Also, the mechanism of the soliton formation is new, resulting from a collective many-body effect.

Regarding the terminology, we have found that in discrete systems the word “solitons” was previously used to describe propagation of wave-like excitations through macroscopic mechanical metamaterials made of discrete interacting elements, see for example:

- B. G. Chen et al. *Proc. Natl Acad. Sci. USA* **111**, 13004-13009 (2014).
- B. Deng et al. *Nat. Commun.* **9**, 3410 (2018).
- Y. Zhang et al. *Nat. Commun.* **10**, 5605 (2019).

We have cited these works in the introduction (page, second column).

The solitary cluster waves do not show dispersion because the nearly barrier-free propagation of clusters requires the solitons to have a particular size for a given wavelength of the potential (and given particle diameter). This size is determined by the force conditions [Eq. (2)], as we explain in detail in the SI.

Reviewer: 2) *Can the authors specify, how they define the size of the cluster? – Specifically, what defines which particles are colored in red in the figures?*

Authors: We describe this point on page 4, column 2 of the manuscript. In the experiments and in the simulations, we identify a cluster as a sequence of neighbouring particles, where the empty gap between two neighbouring particles is smaller than a cut-off distance of $\delta = 5 \times 10^{-3} \lambda$, where λ is the wavelength of the periodic potential. For example, for Fig.1d, $\lambda = 4.65 \mu\text{m}$ and in absence of solitons (left plot with $N = M = 1$), the particles are equispaced at an average distance of $2\pi R/N \sim 4.65 \mu\text{m}$, which corresponds to a free space between the particles of 650 nm. In contrast, when a soliton is present, it is composed by particles that almost touch each other, such that the free space between them is equal or smaller than $\delta = 20\text{nm}$.

Reviewer: 3) *Figure 1a uses dark red, which is later not used anymore. This might be confusing for the reader.*

Authors: In Fig. 1a we have indicated in dark red two particles within a soliton that were occupying the same potential well. Now we have use instead two arrows and have chosen the same colour for all particles forming the soliton. We have also updated the corresponding figure caption.

Reviewer: 4) *I do believe that the paper would benefit tremendously from an improved presentation of Fig. 2 in a more compressed and intuitive way? – There are several redundancies in this figure. Maybe the authors can find some way to compress the figure with a simpler design. Especially Fig. 2a,b were hard to understand (for me).*

Authors: We eliminated all redundancies in Fig. 2, which are described in the supporting information and have reduced the corresponding discussion in the text on the two types of solitons.

Reviewer: 5) *Can the experiment be understood completely classically?*

Authors: The collective motion of particles across the optical potential landscape can be understood from the formation of clusters in classical Brownian dynamics of hard-sphere interacting particles in a periodic potential.

Reviewer: *Minor comments:*

I would give the reason why particle displacement in the radial direction can be neglected in the main part of the text. Additional information can then be given in the methods.

Authors: We point to this fact now by writing on page 2, column 1:

“The generated optical potential confines the particles strongly in the radial direction. Particle displacements in the radial direction have a Gaussian distribution with a mean at the ring radius $R = 20 \mu\text{m}$ and a standard deviation smaller than $0.01R = 0.2 \mu\text{m}$, i.e. they are negligible compared to the particle motion along the ring.”

The distributions of particle displacements in the radial direction from the average position of the particles were determined by us for $N = M$, $N = M+1$ and $N = M+2$. These distributions are Gaussians with standard deviations smaller than $0.2 \mu\text{m}$. The image below shows the distribution for $N = M+1$ as an example, where the red curve marks the Gaussian. Hence, the particle motion is effectively one-dimensional along the ring.

Reviewer: *I would refrain from using the wording “phase difference” to describe the distance, as the first terminology is often used in the quantum mechanical context with a completely different meaning.*

Authors: We have followed the Reviewer’s suggestion and have written “angle distance” instead of “phase difference” throughout the text.

We provide a marked copy of the revised manuscript, where the changes described above are marked in blue.

Response to reviewer comments, second round

Reviewer #1 (Remarks to the Author):

I agree with the authors, that typically the FK model considers harmonic interactions between the particles which is clearly not the case in the experiments discussed here. However, such difference lead to my opinion not to an entirely new behavior compared to FK.

As already written in my first report, my main point was, that the authors did not make any connection to the FK model at all. This issue is now fixed in the revised version. With the changes in the revised version, it becomes now clear how and where differences and similarities with the FK model exist and this will help readers very much to appreciate the work of the authors. I am happy to support publication in Nat. Comm.

Reviewers #2 and #3, who co-reviewed (Remarks to the Author):

In the previous version of the manuscript, the reviewers raised a number of technical comments and also raised the question of whether the results are of sufficient novelty and impact to be appropriate for Nature Communications. The authors have made numerous changes to the manuscript in response to the technical content and have significantly clarified the presentation. They have also more explicitly referenced the previous work by a subgroup of the authors in which some of the theoretical results have already been published.

In its current form, I find that the technical content of the manuscript does not require further adjustment, but I remain unconvinced by the authors' claims that their observation of solitons in this system reaches the level of novelty and impact expected for this journal. In particular, the authors claim that the fact that their system is composed of hard-sphere interacting particles makes it fall outside the realm of the well-known Frenkel-Kontorova (FK) model. This is an overly sweeping claim and there are counterexamples in the literature. For example, in B. van der Meer et al, Phys. Rev. Lett. 121, 258001 (2018), "voidion" excitations in a system of hard cubes are mapped to the sine-Gordon equation, the continuum limit of FK. Thus I recommend that the paper is best suited for a more specialized journal.

Reviewer #4 (Remarks to the Author):

The authors have made a great effort to answer the reviewers questions and have improved the presentation of their results tremendously. Especially the figures convey the their story/results much more succinctly now (in my opinion).

I believe the topic is interesting and suitable for the Nat. Com. community and therefore recommend publication.

Response Letter to Reviewers

We thank all Reviewers for their thorough reading of our manuscript and their suggestions and criticisms. Our point-by-point responses and the corresponding changes to the manuscript are described below.

Response to Reviewer #1

Reviewer: *I agree with the authors, that typically the FK model considers harmonic interactions between the particles which is clearly not the case in the experiments discussed here. However, such difference lead to my opinion not to an entirely new behavior compared to FK.*

As already written in my first report, my main point was, that the authors did not make any connection to the FK model at all. This issue is now fixed in the revised version. With the changes in the revised version, it becomes now clear how and where differences and similarities with the FK model exist and this will help readers very much to appreciate the work of the authors. I am happy to support publication in Nat. Comm.

Authors: We thank the Reviewer for supporting publication of our manuscript.

Response to Reviewers #2 and #3

Reviewers: *In the previous version of the manuscript, the reviewers raised a number of technical comments and also raised the question of whether the results are of sufficient novelty and impact to be appropriate for Nature Communications. The authors have made numerous changes to the manuscript in response to the technical content and have significantly clarified the presentation. They have also more explicitly referenced the previous work by a subgroup of the authors in which some of the theoretical results have already been published.*

Authors: We thank the Reviewer(s) for appreciating our changes made in the revised manuscript to clarify its presentation.

In its current form, I find that the technical content of the manuscript does not require further adjustment, but I remain unconvinced by the authors' claims that their observation of solitons in this system reaches the level of novelty and impact expected for this journal. In particular, the authors claim that the fact that their system is composed of hard-sphere interacting particles makes it fall outside the realm of the well-known Frenkel-Kontorova (FK) model. This is an overly sweeping claim and there are counterexamples in the literature. For example, in B. van der Meer et al, Phys. Rev. Lett. 121, 258001 (2018), "voidion" excitations in a system of hard cubes are mapped to the sine-Gordon equation, the continuum limit of FK. Thus I recommend that the paper is best suited for a more specialized journal.

Authors: We never argued that the hard-sphere interaction make our system "fall outside the realm of the well-known Frenkel-Kontorova (FK) model". With respect to the FK

model, we state in the manuscript that the mechanism leading to the soliton formation is different from the FK model (see second paragraph of Discussion on page 6): “*The mechanism behind the kink formation in the FK model is, however, fundamentally different from the mechanism of soliton formation reported here, which is given by the conditions in Eq. (2) between the external forces.*” The conditions on the external forces responsible for the soliton formation and propagation have, to the best of our knowledge, no counterpart in the FK model.

The work by van der Meer *et al.*, *Phys. Rev. Lett.* **121**, 258001 (2018), is a simulation study, showing that extended vacancy defects (“voidions”) in a crystal of hard cubes (and crystals formed by rhombic prisms and particles interacting via a soft isotropic pair potential) show similarities to crowdions in atomic systems. This similarity is demonstrated by comparing simulated averaged particle displacements (strain) fields in the vacancy defects with analytical results for solitons found in solutions of the sine-Gordon equation, which corresponds to a continuum limit of the FK model. The motion of the vacancy defects is diffusive and described by a persistent random walk.

By contrast, in our experimental system, the driven Brownian motion of spherical particles in a periodic optical potential leads to solitons in form of propagating waves of particle clusters. Their persistent motion is explained by analytical derivations and confirmed by simulations.

Thus, we respectfully disagree with the Reviewer’s opinion that our work deserves publication in a more specialized journal.

We have added the citation to the publication by van der Meer *et al.* when referring to the FK model (see Ref. [53]), and three further citations referring to the formation of crowdions and voidions (see Refs. [50], [51], and [52]).

Response to Reviewer #4

Reviewer: *The authors have made a great effort to answer the reviewers’ questions and have improved the presentation of their results tremendously. Especially the figures convey their story/results much more succinctly now (in my opinion).*

I believe the topic is interesting and suitable for the Nat. Com. community and therefore recommend publication.

Authors: We thank the Reviewer for this positive assessment of our work and for supporting the publication of our manuscript in *Nature Communications*.